

# Sim-DRS: a similarity-based dynamic resource scheduling algorithm for microservice-based web systems

Yiren Li[1,2], Tieke Li[1], Pei Shen[2], Liang Hao[2], Wenjing Liu[3], Shuai Wang[3], Yufei Song[3] and Liang Bao[3]

[1] School of Economics and Management, University of Science and Technology Beijing, Beijing, China
[2] HBIS Group Co., Ltd., Shijiazhuang, China
[3] School of Computer Science and Technology, Xidian University, Xi'an, China

## ABSTRACT

Microservice-based Web Systems (MWS), which provide a fundamental infrastructure for constructing large-scale cloud-based Web applications, are designed as a set of independent, small and modular microservices implementing individual tasks and communicating with messages. This microservice-based architecture offers great application scalability, but meanwhile incurs complex and reactive autoscaling actions that are performed dynamically and periodically based on current workloads. However, this problem has thus far remained largely unexplored. In this paper, we formulate a problem of Dynamic Resource Scheduling for Microservice-based Web Systems (DRS-MWS) and propose a similarity-based heuristic scheduling algorithm that aims to quickly find viable scheduling schemes by utilizing solutions to similar problems. The performance superiority of the proposed scheduling solution in comparison with three state-of-the-art algorithms is illustrated by experimental results generated through a well-known microservice benchmark on disparate computing nodes in public clouds.

# INTRODUCTION

As a new computing paradigm, Microservices have been increasingly developed and adopted for various applications in the past years. Driven by this trend, Microservice-based Web Systems (MWS), which have emerged as a prevalent model for distributed computing, are designed as a set of independent, small and modular microservices implementing individual tasks and communicating with messages. MWS facilitate fast delivery and convenient update of web-based applications, with auto-scalability for the provisioning of virtualized resources, which could be schedule-based, event-triggered, or threshold value-based (*Guerrero, Lera & Juiz, 2018b*). These condition-triggered auto-scaling mechanisms are *reactive*, meaning that the decision on autoscaling actions is made dynamically and periodically based on the current workload. Another important characteristic of MWS is that they are often deployed in a *multi-cloud environment* (*Fazio et al., 2016*); this is because modern Web-based applications are often across

Corresponding author
Wenjing Liu,
wjliu97@stu.xidian.edu.cn

organizational boundaries. For example, a common on-line shopping scenario typically involves an e-commence company, a manufacturer, and a bank, all of which host their information systems on their own cloud.

Applications deployed on the cloud need to meet performance requirements, such as response time, and also need to reduce the cost of cloud resource usage. Generally, Web-based application service requests are submitted by users and run in cloud service providers' cluster environments. Resource scheduling requires instance scheduling and auto-scaling within a restricted time limit to determine the number of instances for each microservice and decide how to deploy each instance to the appropriate VM, achieving the goal of minimizing resource consumption and best service performance as well as meeting user dynamic requests. However, the problem of Dynamic Resource Scheduling for Microservice-based Web Systems (DRS-MWS) is extremely challenging mainly due to the following factors:

- **NP-hard**. The instance scheduling problem in cloud environment has long been proved to be a typical NP-hard problem (*Salleh & Zomaya, 2012*). Due to the dynamic arrival of microservice requests, resource scheduling algorithms are required to adapt to rapidly changing requirements and environments. Under strict time constraints, it becomes more difficult to find a approximate optimal solution.
- **Multi-objective optimization**. DRS-WMS is a complex multi-objective optimization problem. In this problem, the solution may involve many conflicting and influencing objectives, and the researcher should obtain the best possible optimization of these objectives simultaneously. For example, resource scheduling algorithms need to provide sufficient service performance for users while maintaining system robustness to prevent system failure.

Existing methods for resource scheduling for MWS include rule-based, heuristic, and learning-based, as discussed in the Related Work section. Among them, evolutionary algorithms (EA), which are heuristic in nature, have recently received a great deal of attention. EA-based approaches are effective in solving complex microservice scheduling problem (*Fazio et al., 2016*), but suffer from inefficiency and thus fail to satisfy the requirement for reactive and dynamic scheduling. This is mainly because they do not consider *a priori* knowledge about the solution, and often start from a randomly generated initial population. Due to the dynamic nature and the hard time constraint in the microservice scheduling problem, starting with a random population may lead to non-convergence and hence jeopardize the exploration for better solutions (*Wang et al., 2009*).

To address the challenge of dynamic scheduling, we propose Similarity-based Dynamic Resource Scheduling, referred to as Sim-DRS, which aims to quickly find viable scheduling schemes for MWS under a certain time constraint. We tackle the problem of Dynamic Resource Scheduling for Microservice-based Web Systems based on one key hypothesis that **solutions to similar problems often share certain structures**. Therefore, instead of starting from a random population indiscriminately at each initial iteration in a typical EA approach, we focus on finding solutions to similar problems as part of the

initial population to improve the quality of population initialization. This strategy has been shown to be powerful for producing better solutions in the literature (*Rahnamayan, Tizhoosh & Salama, 2007*; *Wang et al., 2009*) and in our practical experiments.

In summary, our work makes the following contributions to the field:

- We formulate DRS-MWS as a *combinatorial optimization* problem.
- We propose Sim-DRS to solve DRS-MWS, which finds promising scheduling schemes by directly utilizing viable solutions to similar problems, hence obviating the need for a fresh start.
- We evaluate the performance of Sim-DRS through extensive experiments using the well-known microservice benchmark named TeaStore under different scheduling time constraints. We show that Sim-DRS outperforms three state-of-the-art scheduling algorithms by 9.70–42.77% in terms of three objectives, and achieves more significant improvements under stricter time constraints.

The remainder of this paper is organized as follows. The "Related Work" section surveys related work and the Problem Statement presents the analytical models of a microservice-based application and formulates the DRS-MWS problem. The "Resource Scheduling Algorithm" section designs Sim-DRS, a dynamic resource scheduling algorithm based on similarity. The "Experiments" section describes the experimental setup and evaluates the scheduling algorithm. In the end, the "Conclusion and Future Work" section presents a discussion of our approach and a sketch of future work.

# RELATED WORK

Resource scheduling for MWS is an active research topic (*Fazio et al., 2016*) and has received a great deal of attention from both industry and academia. Previous studies can be classified into three categories: rule-based, heuristic, and learning-based approaches, as discussed below.

## Rule-based approach

*Yan, Chen & Shuo (2017)* proposed an elastically scalable strategy based on container resource prediction and message queue mapping to reduce the delay of service provisioning. *Leitner, Cito & Stöckli (2016)* proposed a graph-based model for the deployment cost of microservices, which can be used to model the total deployment cost depending on the call patterns between microservices. *Magalhaes, Rech & Moraes (2017)* proposed a scheduling architecture consisting of a Web server powered by a soft real-time scheduling engine. *Gabbrielli et al. (2016)* proposed JRO (Jolie Redeployment Optimiser) tool to generate a suggested SOA (service-oriented architecture) configuration from a partial and abstract description of a target application. *Filip et al. (2018)* proposed a mathematical formulation for describing an architecture that includes heterogeneous machines to handle different microservices. *Zheng et al. (2019)* presented SmartVM, a business Service-Level-Agreement (SLA)-aware, microservice-centric deployment framework to handle traffic spikes in a cost-efficient manner. *Fard, Prodan & Wolf (2020)* proposed a general microservice scheduling mechanism and modeled the scheduling

problem as a complex variant of the knapsack problem, which can be expanded for various resource requests in queues and solved by multi-objective optimization methods. *Mirhosseini et al. (2020)* developed a scheduling framework called Q-Zilla from the perspective of solving the end-to-end queue delay, and the SQD-SITA scheduling algorithm was proposed to minimize the delay caused by microservice distribution.

Rule-based approaches are straightforward and are efficient in simple environments. The scheduling problem can be solved by constructing rules through domain knowledge, using software architecture and simple data modeling theory, which is effective in an environment that meets certain assumptions. However, they rely heavily on prior domain knowledge, have a low degree of mathematical abstraction, and may be labor-intensive, imprecise, and have poor results in high variability scenarios.

## Heuristic approach

*Li et al. (2018)* proposed a prediction model for microservice relevance using optimized artificial bee colony algorithm (OABC). Their model takes into account the cluster load and service performance, and has a good convergence rate. *Stévant, Pazat & Blanc (2018)* used a particle swarm optimization to find the best placement based on the performance of microservices evaluated by the model on different devices to achieve the fastest response time. *Guerrero, Lera & Juiz (2018b)* presented an NSGA-II algorithm to reduce service cost, microservice repair time, and microservice network latency overhead. *Adhikari & Srirama (2019)* used an accelerated particle swarm optimization (APSO) technique to minimize the overall energy consumption and computational time of tasks with efficient resource utilization with minimum delay. *Lin et al. (2019)* proposed an ant colony algorithm that considers not only the utilization of computing and storage resources but also the number of microservice requests and the failure rate of physical nodes. *Guerrero, Lera & Juiz (2018a)* proposed an NSGA-II-based approach to optimize system provisioning, system performance, system failure, and network overhead simultaneously. *Bhamare et al. (2017)* presented a fair weighted affinity-based scheduling heuristic to reconsider link loads and network delays while minimizing the total turnaround time and the total traffic generated. *Lin et al. (2019)* used an ant colony algorithm to solve the scheduling problem. It considered not only the computing and storage resource utilization of physical nodes, but also the number of microservice requests and failure rates of the nodes, and combined multi-objective heuristic information to improve the probability of choosing optimal path.

These approaches generally abstract resources scheduling into an optimization problem through appropriate modeling methods and solve the problem in a certain neighborhood. And heuristic approaches have been proven to be efficient in finding good scheduling solutions in a high-dimensional space (*Guerrero, Lera & Juiz, 2018a*), especially under the circumstances of balancing many conflicting objectives. However, they suffer from low performance and search from a random state, which lead to noneffective use of *a priori* knowledge of existing good solutions.

### Learning-based approach

*Alipour & Liu (2017)* presented a microservice architecture that adaptively monitors the workload of a microservice and schedules multiple machine learning models to learn the workload pattern online and predict the microservice's workload classification at runtime. *Nguyen & Nahrstedt (2017)* proposed MONAD, a self-adaptive microservice-based infrastructure for heterogeneous scientific workflows. MONAD contains a feedback control-based resource adaptation approach to generate resource allocation decisions without any knowledge of workflow structures in advance. *Gu et al. (2021)* proposed a dynamic adaptive learning scheduling algorithm to intelligently sorts, allocates, monitors, and adjusts microservice instances online. *Yan et al. (2021)* used the neural network and attention mechanism in deep learning to optimize the passive elastic scaling mechanism of the cloud platform and the active elastic mechanism of microservices by accurately predicting the load of microservices, and finally realized the automatic scheduling of working nodes. *Lv, Wei & Yu (2019)* used machine learning methods in resource scheduling in microservice architecture, pre-trained a random forest regression model to predict the requirements for the microservice in the next time window based on the current unload pressure, and the number of instances and their locations were adjusted to balance the system pressure.

The learning-based solutions are still in their infancy. The main advantage of these approaches is that they can generate scheduling decisions adaptively and automatically, without any human intervention. However, these approaches require a considerable number of samples to build a reasonable decision model for a microservice system (*Alipour & Liu, 2017*). The high demand of samples is always challenging because only a very limited set of samples can be acquired during a short time period for resource scheduling in production systems.

## PROBLEM STATEMENT

We study a Dynamic Resource Scheduling problem for Microservice-based Web Systems, referred to as *DRS-MWS*. As is shown in Fig. 1, a microservice-based Web system is often deployed in a multi-cloud environment consisting of a set of interconnected virtual machines. This system acts as a real-time streaming data pipeline that delivers data and messages to microservices. Users can send requests to microservices once deployed according to their own requirements. Each type of microservice provides a unique function, and multiple microservices collectively constitute an integrated service system. Given a batch of dynamic requests at runtime, the goal of DRS-MWS is to find an optimal provisioning policy to improve the system's service quality and robustness while ensuring the high quality.

### System components

Specifically, DRS-MWS has the following components.

#### *Microservice*

A microservice-based Web System (MWS) contains many kinds of microservice, and each type provides a specific functionality, which can be modeled as a three tuple

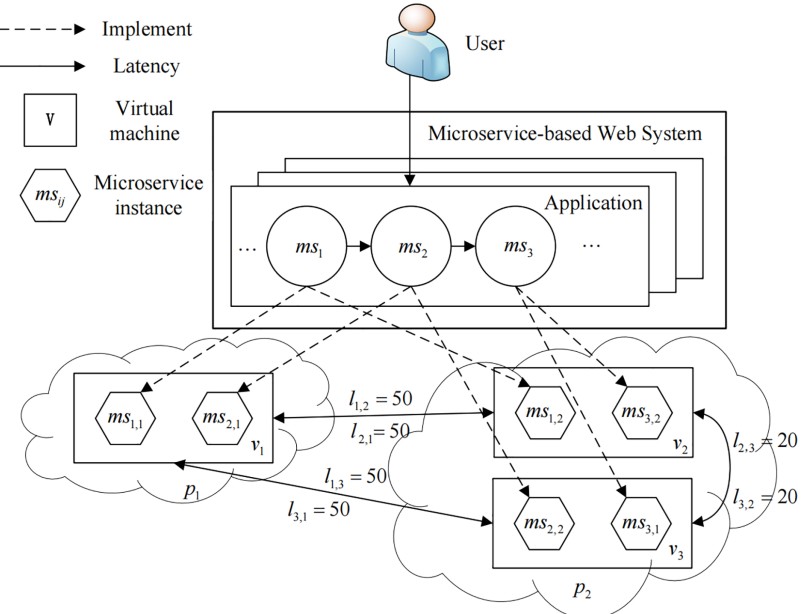

**Figure 1 An overview of the *DRS-MWS* problem.**

$ms = \langle C_{ms}, R_{ms}, g \rangle$, where $C_{ms}$ and $R_{ms}$ represent the normalized CPU and memory resource demand for deploying $ms$ on VMs, respectively; $g$ denotes the full capacity of $ms$ to achieve its functionality.

### Application

Each application, which consists of many microservices and represents a useful business functionality, can implement a corresponding type of requests from users. In this paper, we model an application as a directed acyclic graph (DAG) A = $\langle V_A, E_A \rangle$, where vertices $V_A$ represent a set of $m$ microservices $V_A = \{ms_1, ms_2, \cdots, ms_m\}$. The execution dependency between a pair of adjacent microservices $ms_i$ and $ms_j$ is denoted by a directed edge $(ms_i, ms_j) \in E_A$ between them.

### Workload

By definition, workload often represents the requests to an application from different users at a given time point $t$. For clarity, we use *microservice* workload (short for workload) here instead of application workload because we need to track the details of microservice requests in the *DRS-MWS* problem. Given a set of $m$ microservices, we model workload as $W^{(t)} = \{w_1^{(t)}, w_2^{(t)}, \cdots, w_m^{(t)}\}$, where each integer $w_i^{(t)}$ denotes the total number of user requests for $ms_i$ at $t$, which can reflect the duration the requests are queued in $ms_i$ waiting to be executed.

### Running environment

We consider a set of cloud providers $P = \{p_1, p_2, \cdots\}$ hosting $n$ virtual machines (VMs). Typically, different types of VMs with different computing capabilities characterized by the number of virtual CPU cores, CPU frequency, RAM and disk size are provisioned to satisfy

different application needs. For simplicity, we define a normalized scalar $p = \langle C_{vm}, R_{vm} \rangle$ to describe such computing capability for a given VM, where $C_{vm}$ and $R_{vm}$ represents the virtual CPUs and memory allocated to VM. To model the communication time or latency overhead among different cloud providers, we define a matrix $L_{n \times n}$ where the element $l_{i,j}$ represents the latency between the VM $v_i$ and $v_j$. Note that we assume that the latency within the same VM is negligible, *i.e.* $l_{i,i} = 0$, and the latency within the different cloud providers is larger than in the same provider. As shown in Fig. 1, there are three VMs ($v_1$, $v_2$, and $v_3$), and the latency can be expressed as:

$$L = \begin{bmatrix} 0 & 50 & 50 \\ 50 & 0 & 20 \\ 50 & 20 & 0 \end{bmatrix}. \tag{1}$$

### *Microservice Instance*

To implement a business function, each microservice needs to be deployed in a container to create a microservice instance. Without loss of generality, we follow the popular "service instance per container" (*Richardson, 2020*) deployment pattern in this paper and use the term *microservice instance* to denote both the software and the container infrastructure of a specific microservice. Specifically, we use $ms_{i,j}$ to represent the $j$-th instance of microservice $ms_i$. As illustrated in Fig. 1, function $f_1$ is mapped to $ms_{1,1}$ and $ms_{1,2}$, which means the 1-st instance of microservice $ms_1$ and the 1-st instance of microservice $ms_2$ work cooperatively to implement $f_1$.

## Optimization objectives

At any time point $t$, we wish to optimize three objectives: (i) the resource consumption ($C^{(t)}$) for supporting users' requests; (ii) the system jitter ($J^{(t)}$) due to the deploying adjustment of the microservice instances; and (iii) the invocation expense ($E^{(t)}$) for calling different microservice instances along the microservice invocation chain of an application. It is worth mentioning that we treat a microservices as a black-box function, and the latency of queueing and executing requests within the microservice is not considered in this paper.

To achieve this optimization, we define our *resource scheduling* first by managing the number of instances for each microservice and then deciding how to deploy each microservice instance to an appropriate VM. More specifically, suppose that an MWS has $n$ VMs and $m$ different microservices, and a workload $W^{(t)} = \{w_1^{(t)}, w_2^{(t)}, \cdots, w_m^{(t)}\}$ is generated at a time point $t$. We define a matrix $S_{m \times n}$ to denote our scheduling decision, where each element $s_{ij} \in S$ indicates the number of instances for microservice $ms_i$ that will be deployed to VM $v_j$, and for example, we have three VMs ($v_1$, $v_2$, and $v_3$) and three microservices ($ms_1$, $ms_2$, and $ms_3$) in Fig. 1, and the current resource scheduling at time point $t$ can be defined as:

$$S^{(t)} = \begin{bmatrix} 1 & 1 & 0 \\ 1 & 0 & 1 \\ 0 & 1 & 1 \end{bmatrix}. \tag{2}$$

**Resource consumption**, denoted by $C^{(t)}$, is measured as the sum of the resource demands for deploying each microservice instance on its target VM:

$$C^{(t)} = \sum_{i=1}^{m} \sum_{j=1}^{n} (s_{i,j}^{(t)} \cdot ms_i.r), \qquad (3)$$

where $s_{i,j}^{(t)}$ denotes the number of microservice instances $ms_i$ deployed on VM $v_j$, and $ms_i.r$ represents the normalized resource demand for deploying $ms_i$ on a VM.

**System jitter**, denoted by $J^{(t)}$, is an important performance metric used to measure the robustness of the system at time point $t$, which can be further defined as the change degree of microservice instances' deployment for an MWS environment. For example, given any two continuous time points $t$ and $t-1$, $J^{(t)}$ results from subtracting scheduling decisions $S^{(t)}$ and $S^{(t-1)}$:

$$J^{(t)} = \sum_{i=1}^{m} \sum_{j=1}^{n} \left| s_{i,j}^{(t)} - s_{i,j}^{(t-1)} \right|. \qquad (4)$$

**Invocation expense**, denoted by $E^{(t)}$, is defined as the *associated cost* for considering both the microservice invocation chains of applications and the latency overhead among different cloud providers. Because the latency has been defined in the previous section, we need to define the former as microservice invocation expense.

Given a set of applications $A = \{A_1, A_2, \cdots\}$ in an MWS, we define the correlation, denoted as $cor(ms_i, ms_j)$, between any two microservices $ms_i$ and $ms_j$ as:

$$cor(ms_i, ms_j) = \begin{cases} 1 & \text{if } (ms_i, ms_j) \in E_{A_k}, \forall A_k \in A \\ 0 & \text{otherwise} \end{cases}, \qquad (5)$$

where $EA_k$ represents the microservices required to implement $A_k$.

The microservice invocation distance, denoted by a matrix $D_{m \times m}$, is thus defined to indicate the alienation between any two microservices, and each element $d_{i,j}$ in $D$ is defined as:

$$d_{i,j} = \begin{cases} e^{-cor(ms_i, ms_j)} & \text{if } i \neq j \\ 0 & \text{otherwise.} \end{cases} \qquad (6)$$

As shown in Fig. 1, there is only one application $A = \{ms_1, ms_2, ms_3\}$ that belongs to the MWS, and the microservice distance $D$ is thus represented as:

$$D = \begin{bmatrix} 0 & e^{-1} & 1 \\ 1 & 0 & e^{-1} \\ 1 & 1 & 0 \end{bmatrix}. \qquad (7)$$

Based on the latency $L_{n \times n}$ among different VMs, the scheduling decision $S^{(t)}{}_{m \times n}$ at time point $t$, and the microservice distance $D_{m \times m}$, we now define the invocation expense $E^{(t)}$ as a scalar value:

$$E^{(t)} = U \cdot (D \cdot S^{(t)} \cdot L) \cdot U^T, \qquad (8)$$

where $U_{1 \times n}$ is an auxiliary matrix whose elements are all equal to 1. For example, as shown in Fig. 1, given $L$ defined in Eq. (1), $S^{(t)}$ defined in Eq. (2), and $D$ defined in Eq. (7), we have:

$$E^{(t)} = \begin{bmatrix} 1 & 1 & 1 \end{bmatrix} \left( \begin{bmatrix} 0 & e^{-1} & 1 \\ 1 & 0 & e^{-1} \\ 1 & 1 & 0 \end{bmatrix} \begin{bmatrix} 1 & 1 & 0 \\ 0 & 1 & 1 \\ 0 & 1 & 1 \end{bmatrix} \begin{bmatrix} 0 & 50 & 50 \\ 50 & 0 & 20 \\ 50 & 20 & 0 \end{bmatrix} \right) \begin{bmatrix} 1 \\ 1 \\ 1 \end{bmatrix} = 280e^{-1} + 520 \quad (9)$$

## Problem formulation

We formally define *DRS-MWS* as a three-objective optimization problem:

$$\begin{cases} \min\limits_{\forall S^{(t)}} C^{(t)}(A, MS, V, S^{(t)}) \\ \min\limits_{\forall S^{(t)}} J^{(t)}(A, MS, V, S^{(t)}) \\ \min\limits_{\forall S^{(t)}} E^{(t)}(A, MS, V, S^{(t)}) \end{cases} \quad (10)$$

$$s.t. \quad \text{scheduling time} \leq \gamma \cdot \Delta t \quad (11)$$

$$\sum_{j=1}^{m} (s_{i,j}^{(t)} \cdot ms_j.r) \leq \delta \cdot v_i.r \quad \forall t, i = 1, 2, \cdots, n \quad (12)$$

$$\sum_{j=1}^{n} s_{i,j}^{(t)} = \lceil \frac{w_i}{ms_i.g} \rceil^{(t)} \quad \forall t, i = 1, 2, \cdots, m \quad (13)$$

where Eq. (10) states that at any time point $t$, given an MWS consisting of a set of applications ($A$), a set of microservices ($MS$), and a set of VMs ($V$), the goal of *DRS-MWS* is to find a resource scheduling policy $S$ among all valid policies under workload $W^{(t)}$ to minimize the resource consumption ($C^{(t)}$), system jitter ($J^{(t)}$), and invocation expense ($E^{(t)}$) simultaneously. The constraint Eq. (11) is that any solution to the problem must terminate after a $\gamma \cdot \Delta t$ amount of time. The constraint Eq. (12) states that the resource consumption of any VM for deploying microservices must not exceed a certain proportion of its total resource capacity. Finally, the constraint Eq. (13) states that the workload on each microservice needs to be served appropriately.

## RESOURCE SCHEDULING ALGORITHM

In this section, we introduce Sim-DRS—a similarity-based dynamic resource scheduling algorithm to solve *DRS-MWS*. Its key idea is to accelerate the convergence of the scheduling algorithm by adopting previously-known good solutions as the initial population whose optimization situation is similar to the current one. We first analyze existing scheduling algorithms and their limitations. We then present similarity estimation, which determines an appropriate initial population for the scheduling algorithm at each time point. Finally, we discuss the details of the Sim-DRS algorithm.

### Motivation

Evolutionary algorithms (EA) are among the most popular for solving the *DRS-MWS* problem (*Guerrero, Lera & Juiz, 2018a*). EA implementation requires a definition of the

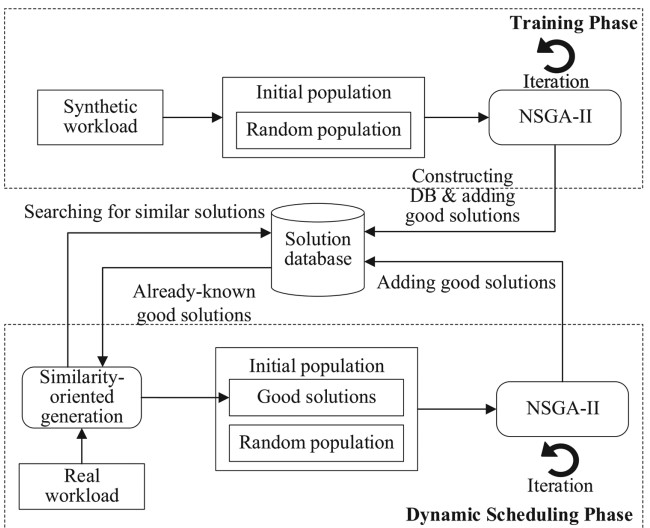

**Figure 2  An overview of the proposed Sim-DRS approach.**

solution and the construction of several technical components including initial population, genetic operators such as crossover and mutation, fitness function, selection operator, offspring generation, and execution parameterization (*Mitchell, 1998*). Most of the previous approaches assume that no *a prior* information about the solution is available, and often use a random initialization method to generate candidate solutions (*i.e.*, the initial population). According to (*Rahnamayan, Tizhoosh & Salama, 2007*), population initialization is a crucial step in evolutionary algorithms because it can affect the convergence speed and also the quality of the final solution. Due to the dynamics and the hard time constraint in the *DRS-MWS* problem, previous studies have shown that adopting a random initial population often leads to non-convergence in the optimization process and eventually a low-quality solution (*Wang et al., 2009*).

   Instead of using a complete randomization technique for initial population generation, we attempt to leverage existing solutions to similar problems to compose the initial population. The rationale behind this idea is that good solutions to similar problems may share some common structures. Figure 2 illustrates the optimization process of Sim-DRS, which contains two phases: offline training and online scheduling. In the training phase, it first generates a set of synthetic workloads, and then applies a Non-dominated Sorting Genetic Algorithm II (NSGA-II) to reach the pareto optimality for three optimization objectives mentioned in Eq. (10). It constructs a database of good solutions by adding such one-to-one <workload-solution> pairs. In the scheduling phase, given a real workload $W^{(t)}$ generated by users at each time point $t$, Sim-DRS first applies a similarity-based algorithm to generate an initial population mixed with random and previously-known similar solutions from the database by calculating the similarities between $W^{(t)}$ and existing synthetic workloads. It then uses a standard NSGA-II algorithm to optimize three objectives defined in Eq. (10) and select the best one.

---

| **Algorithm 1** Solution database construction algorithm: SDC(W, k). |
|---|

**Input:** $W = \{W_1, W_2, \cdots, W_n\}$: the workload set consisting of $n$ synthetic workloads; $k$: the number of clusters.

**Output:** $SD$: the solution database containing $k$ groups.

1: $SD \leftarrow \varnothing$;

2: **for** $i = 1{:}n$ **do**

3:      Randomly generate an initial population with $p$ different solutions;

4:      Find the optimal solution $S_i$ for $W_i$ using an NSGA-II algorithm;

5:      $SD \leftarrow (W_i, S_i)$;

6: **end for**

7: $SD \leftarrow k\text{-}means(SD, k)$;

8: **return** $SD$;

---

## Solution database construction

To measure the correlation between any two workloads $W_i = \{w_1^i, w_2^i, \cdots, w_m^i\}$ and $W_j = \{w_1^j, w_2^j, \cdots, w_m^j\}$, we first define *workload similarity* (or *similarity* in short):

$$\Gamma(W_i, W_j) = e^{-\sqrt{\sum_{k=1}^{m}(w_k^i - w_k^j)^2}}. \tag{14}$$

As shown is Fig. 2, we need to construct a solution database containing previously-known good solutions for different workloads in the training phase. The *solution database construction* (SDC) algorithm is provided in Algorithm 1.

As shown in Algorithm 1, for every workload $W_i$ in the synthetic workload set $W$, we randomly generate $p$ different solutions as the initial population (line 3), and then apply a Non-dominated Sorting Genetic Algorithm II (NSGA-II) to find the optimal solution (line 4). The <workload-solution> pair is added to the database as a known fact (line 5). Note that the training phase is conducted offline, which allows an extensive execution of the NSGA-II algorithm without any time constraint. Based on the definition of similarity, we then apply a $k$-Means clustering algorithm to these pairs by clustering the workloads and generate $k$ different groups, where $k$, which is often designated based on an empirical study, is used to characterize different workload patterns (line 7).

Figure 3 illustrates an example of the solution database containing three groups of workloads $G_1$, $G_2$, and $G_3$, each of which consists of three workloads. Three centers, namely $W_3$, $W_5$, and $W_7$ in this example, represent their clusters, respectively.

## Similarity-oriented initial population generation

Given a workload $W^{(t)}$ generated by users at each time point $t$ during the dynamic scheduling phase, the similarity-based initial population generation algorithm in our Sim-DRS approach aims to find good solutions from the solution database according to the distance measurement, as shown in Algorithm 2.

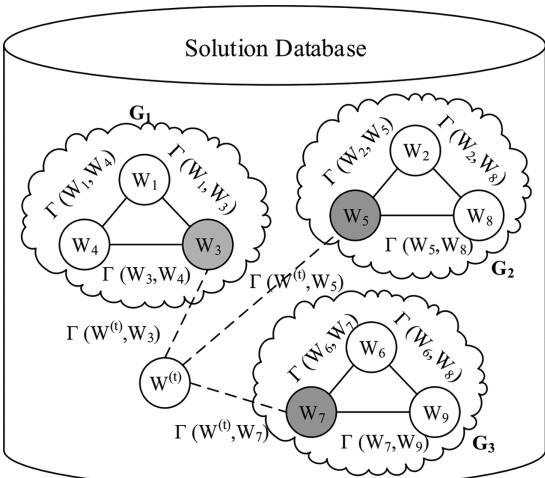

**Figure 3  An example of the solution database.**

---

**Algorithm 2  Similarity-oriented initial population generation algorithm: GenSimPop(W^(t), SD, p).**

**Input:** $W^{(t)}$: the real workload arriving at time point $t$; $SD$: the solution database; $p$: the number of desired individuals in the set of good solutions.

**Output:** $GS$: the set of good solutions for composing the initial population.

1: $GS \leftarrow \varnothing$;

2: $G^* \leftarrow \underset{G \in SD}{\arg\max} \, \Gamma(W^{(t)}, W_G^c)$;

3: Construct a roulette wheel selection process $PS$ with $G^\star$;

4: **for** $i = 1: p$ **do**

5:      $(W_i, S_i) \leftarrow RWS(G^\star, PS)$;

6:      **for** $j = 1: m$ **do**

7:          $diff \leftarrow w^{(t)}_j - w^i_j$;

8:          $adj \leftarrow \lceil \frac{diff}{ms_j.g} \rceil$;

9:          **if** $adj \neq 0$ **then**

10:             $k \leftarrow |adj|$;

11:             **while** $k > 0$ **do**

12:                Randomly select a VM index $x \in [1, n]$;

13:                **if** $s^i_{j,x} + \frac{|adj|}{adj} \geq 0$ **then**

14:                    $s^i_{j,x} \leftarrow s^i_{j,x} + \frac{|adj|}{adj}$;

15:                    $k \leftarrow k - 1$;

16:                **end if**

17:             **end while**

18:          **end if**

19:      **end for**

20:      $GS \leftarrow GS \cup S_i$;

21: **end for**

22: **return** $GS$;

---

The similarity-based initial population generation algorithm begins with searching for the group $G^*$ in the solution database that has the largest similarity with $W^{(t)}$ (line 2), where $W_G^c$ represents the cluster center of any workload group $G$. For example, given a workload $W^{(t)}$ in Fig. 3, we need to calculate $\gamma(W^{(t)}, W_3)$ for $G_1$, $\gamma(W^{(t)}, W_5)$ for $G_2$, and $\gamma(W^{(t)}, W_7)$ for $G_3$, respectively.

Given $G^*$, we construct a roulette wheel selection (RWS) process, a proportional selection strategy which has a similar selection principle as roulette wheel, to construct a population with good solutions (line 3). More specifically, suppose that group $G^*$ contains $k$ different <workload-solution> pairs $G^* = \{a_1, a_2, \cdots, a_k\}$, and an individual pair $a_i = <W_i, S_i>$ has the distance value of $\Gamma(W^{(t)}, W_i)$. Then, the probability for $a_i$ to be selected is:

$$ps(a_i) = \frac{\Gamma(W^{(t)}, W_i)}{\sum\limits_{j=1}^{k} \Gamma(W^{(t)}, W_j)} \quad i = 1, 2, \cdots, k. \tag{15}$$

After obtaining the roulette wheel: $PS = \{ps(a_1), ps(a_2), \cdots, ps(a_k)\}$, we repeatedly select candidate workload-solution pairs $p$ times from $G^*$ using the RWS strategy with $PS$ (line 5).

Once a solution $S_i$ to the workload $W_i$ is selected, we need to adjust it to the current workload $W^{(t)}$ if $W^{(t)} \neq W_i$ (lines 6–19). More specifically, for each microservice $ms_j$, we calculate the difference $diff$ between the workload $w_j^t$ for $W^{(t)}$ and $w_j^i$ for $W_i$ (line 7), and then convert the *workload difference* (*diff*) into the *microservice instance difference* (*adj*) (line 8). To adjust the scheduling decision $s_{j,*}^i$ for $ms_j$ on every VM, we randomly choose a VM indexed by $x$ ($x \in [1, n]$) and add/remove a microservice instance to/from $v_x$ (lines 12–16). This process repeats multiple times until $|adj|$ times of adjustments have been performed successfully (lines 11–17). After each adjustment on $S_i$, we add it to the set of good solutions $GS$ for the initial population (line 20). The algorithm stops after $p$ times of selections and adjustments, and finally returns the set of good solutions $GS$.

### Sim-DRS algorithm

The pseudocode of Sim-DRS is provided in Algorithm 3. Sim-DRS initially generates $h_s$ ($\alpha \cdot populationSize$) number of good solutions using our similarity-based initial population generation algorithm (line 2), and generates $h_r$ (($1 - \alpha$) $\cdot$ populationSize) number of solutions using the standard random algorithm (line 3). Finally, $h_s$ and $h_r$ are merged together to form the initial population (line 4).

Our Sim-DRS approach is based on the Non-dominated Sorting Genetic Algorithm-II (NSGA-II) (*Deb et al., 2002*) (lines 5–25) as introduced in the previous section. The crossover operation randomly exchanges the same number of rows of two individuals to produce new ones (line 12). To avoid the local minimum value and cover a larger solution space, a mutation operator is also used (line 14). Note that in order to satisfy the constraint stated in Eq. (11), we need to randomly adjust the values of an individual after applying the crossover and mutation operations (line 16–17).

---

**Algorithm 3** The Sim-DRS algorithm: SimDRS(W(t), SD).

**Input:** $W^{(t)}$: the real workload arriving at time point $t$; $SD$: the solution database.

**Output:** $S$: the optimal scheduling strategy.

1: Initialize populationSize, generationNumber, mutationProb;

2: $h_s \leftarrow$ GenSimPop($W^{(t)}$, $SD$, $\alpha \cdot$ populationSize);

3: $h_r \leftarrow$ GenRandomPop(($1 - \alpha$) $\cdot$ populationSize);

4: $h \leftarrow h_s + h_r$;

5: fitness $\leftarrow$ CalculateFitness($h$);

6: fronts $\leftarrow$ CalculateFronts($h$, fitness);

7: distance $\leftarrow$ CalculateCrowd($h$, fitness, fronts);

8: **for** $i = 1$ : generationNumber **do**

9:      $h_{off} \leftarrow \emptyset$;

10:      **for** $j = 1$ : populationSize **do**

11:         $fa_1, fa_2 \leftarrow$ BinarySelect($h$, fitness, distance);

12:         $ch_1, ch_2 \leftarrow$ Crossover($fa_1$, $fa_2$);

13:         **if** Random() < mutationProb **then**

14:           Mutate($ch_1$, $ch_2$);

15:         **end if**

16:         $ch_1 \leftarrow RandomAdjustment(ch_1)$;

17:         $ch_2 \leftarrow RandomAdjustment(ch_2)$;

18:         $h_{off} \leftarrow h_{off} \cup \{ch_1, ch_2\}$;

19:      **end for**

20:      $h_{off} \leftarrow h_{off} \cup h$;

21: **end for**

22: fitness $\leftarrow$ CalculateFitness($h_{off}$);

23: fronts $\leftarrow$ CalculateFronts($h_{off}$, fitness);

24: distance $\leftarrow$ CalculateCrowd($h_{off}$, fitness, fronts);

25: $h_{off} \leftarrow$ OrderElements($h_{off}$, fronts, distance);

26: $h \leftarrow h_{off}[1 \ldots$ populationSize$]$;

27: $S \leftarrow fronts[1]$;

28: **return** $S$;

---

According to the problem formulation, Sim-DRS considers three objective functions, namely, $C^{(t)}$, $J^{(t)}$, and $E^{(t)}$, as the fitness functions to measure the quality of a solution (lines 5 and 22). It sorts the solutions at Pareto optimal front levels, and all the solutions in the same front level are ordered by the crowding distance (lines 22–25). Once all the solutions are sorted, a binary tournament selection operator is applied over the sorted elements (line 11): two solutions are selected randomly, and the first one on the ordered list is finally selected (line 27).

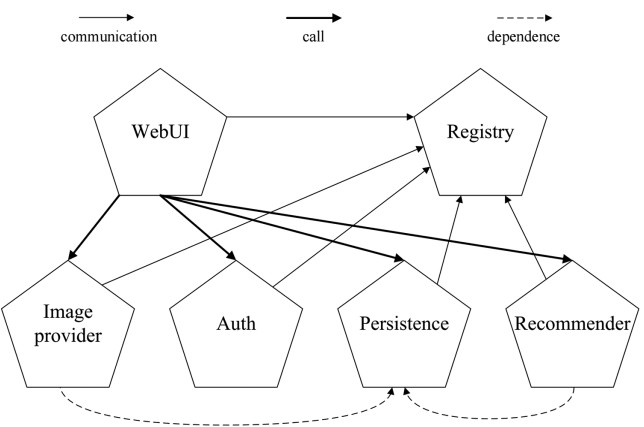

**Figure 4  The architecture of TeaStore benchmark.**

## EXPERIMENTS

We implemented our algorithm and conducted extensive experiments under different testing scenarios. The source code and data can be found in the online public repository (https://github.com/xdbdilab/Sim-DRS). In this section, we first describe our experiment setup, and then present the experimental results to illustrate the efficiency and effectiveness of the proposed approach.

### Experimental setup

#### Benchmark

We choose TeaStore (*von Kistowski et al., 2018*) as our benchmark application to evaluate the performance of different algorithms including Sim-DRS. TeaStore is a state-of-the-art microservice-based test and reference application, and has been widely used for performance evaluation of microservice-based applications. It allows evaluating performance modeling and resource management techniques, and also offers instrumented variants to enable extensive run-time analysis.

As shown in Fig. 4, the TeaStore consists of five distinct services and a *registry* service. All services communicate with the registry for service discovery and load balancing. Additionally, the *WebUI* service issues calls to the *image provider*, *authentication*, *persistence* and *recommender* services. The image provider and recommender are both dependent on the persistence service. All services communicate *via* representational state transfer (RESTful) calls, and are deployed as Web services on the Apache Tomcat Web server.

#### Workload

To characterize user requests to a production microservice-based web system in a daily cycle, we implemented a workload generator using JMeter (https://jmeter.apache.org/) to generate requests according to Poisson and random distribution, and each request corresponds to some kind of application calling the corresponding microservices. For each cycle of workload, we use every algorithm to make scheduling decisions for 20 times under a fixed time constraint. We observed that it usually takes 5 to 10 s for Kubernetes to

**Table 1 Specifications of virtual machines.**

| VMs | CPU cores | Memory size | Disk size |
| --- | --- | --- | --- |
| v1 | 4 | 16 GiB | 30 GB |
| v2 | 4 | 8 GiB | 30 GB |
| v3 | 4 | 8 GiB | 30 GB |
| v4 | 4 | 8 GiB | 30 GB |
| v5 | 8 | 16 GiB | 30 GB |

create a new container or destroy an existing container in our experiment environment, so the time constraint should be smaller then 10 s in order to minimize the influence on MWS. Based on this observation, we set five different time intervals for resource scheduling, *i.e.*, 2 s, 4 s, 6 s, 8 s, and 10 s, in our experiment.

### Execution environment

The experiment was carried out on a cluster of VMs provisioned on public clouds, each of which runs CentOS Linux release 7.6.1810 (core) X86-64. The specifications of these five VMs are provided in Table 1. Specifically, $v_1$ deploys the Nginx gateway and the complete set of TeaStore test benchmarks, including the database and the registry. The other four virtual machines ($v_2$–$v_5$) are used for the deployment of microservice instances. More specifically, $v_2$, $v_3$ and $v_4$ are deployed in one public cloud, and $v_5$ is deployed in the other public cloud.

### Performance metrics

We consider three objectives in our experiments for performance evaluation, namely, resource consumption ($C^{(t)}$), system jitter ($J^{(t)}$), and invocation expense ($E^{(t)}$), as formally defined in Eq. (10). The performance improvement of an algorithm over a baseline algorithm in comparison is defined as:

$$Imp(baseline) = \frac{P - P_{baseline}}{P_{baseline}} \cdot 100\%, \tag{16}$$

where $P_{baseline}$ is the performance of the baseline algorithm, and $P$ is that of the algorithm being evaluated.

For each run in our experiments, every algorithm is executed under the same time constraint and stops once the constraint is met. To ensure consistency, we run each workload five times and calculate the average of these five runs.

### Baseline algorithms and hyperparameters

To evaluate the performance of Sim-DRS, we compare it with three state-of-the-art algorithms, namely, Ant Colony Algorithm (ACO) (*Merkle, Middendorf & Schmeck, 2002*), Particle Swarm Optimization (PSO) (*Kumar & Raza, 2015*), and Non-dominated Sorting Genetic Algorithm II (NSGA-II) (*Deb et al., 2002*). Table 2 summarizes the hyperparameters for each algorithm (including Sim-DRS).

**Table 2 Hyperparameters for each algorithm.**

| Algorithms | Parameter name | Value |
|---|---|---|
| ACO | pheromone volatilization rate | 0.5 |
| | pheromone initial concentration | 700 |
| | pheromone releasing factor | 1 |
| | information heuristic factor | 3 |
| | expectation heuristic factor | 1 |
| | number of ants | 50 |
| PSO | iteration | 50 |
| NSGA-II | mutation probability | 0.3 |
| | cross probability | 0.3 |
| | population | 50 |
| Sim-DRS | mutation probability | 0.3 |
| | cross probability | 0.3 |
| | population | 50 |
| | the number of clusters | 5 |
| | the proportion of good solutions ($\alpha$) | 0.4 |

**Table 3 Objective measurements of different algorithms under Poisson distribution.**

| Objectives | Time (s) | ACO (Imp%) | PSO (Imp%) | NSGA-II (Imp%) | Sim-DRS |
|---|---|---|---|---|---|
| Resource consumption | 2 | 21.534 (10.58%) | 21.890 (12.41%) | 21.660 (11.23%) | 19.474 |
| | 4 | 21.702 (11.67%) | 21.534 (10.75%) | 21.286 (9.47%) | 19.444 |
| | 6 | 21.732 (12.86%) | 21.706 (12.72%) | 21.674 (12.56%) | 19.256 |
| | 8 | 21.256 (10.09%) | 21.396 (10.81%) | 21.218 (9.89%) | 19.308 |
| | 10 | 21.018 (9.33%) | 21.206 (10.31%) | 21.070 (9.60%) | 19.224 |
| System jitter | 2 | 8.80 (51.72%) | 8.80 (51.72%) | 7.60 (31.03%) | 5.80 |
| | 4 | 7.80 (56.00%) | 7.60 (32.00%) | 6.40 (28.00%) | 5.00 |
| | 6 | 5.40 (28.57%) | 6.50 (54.76%) | 6.40 (52.38%) | 4.20 |
| | 8 | 5.80 (38.10%) | 6.00 (42.86%) | 6.20 (47.62%) | 4.20 |
| | 10 | 5.30 (39.47%) | 4.80 (26.32%) | 5.00 (31.58%) | 3.80 |
| Invocation expense | 2 | 20.080 (9.25%) | 20.990 (14.20%) | 20.340 (10.66%) | 18.380 |
| | 4 | 20.330 (9.30%) | 20.380 (9.03%) | 20.730 (11.45%) | 18.600 |
| | 6 | 20.540 (9.61%) | 20.680 (10.35%) | 20.700 (10.46%) | 18.740 |
| | 8 | 20.260 (10.47%) | 20.080 (9.49%) | 20.140 (9.81%) | 18.340 |
| | 10 | 19.580 (9.88%) | 19.570 (9.82%) | 19.890 (11.62%) | 17.820 |

## Experimental results

Given fixed time constraints, we run four different scheduling algorithms independently. Table 3 tabulates the objective values when the workload subjects to the Poisson distribution. As expected, Sim-DRS has better performance than the three comparison algorithms in terms of three objectives. Specifically, in terms of resource consumption, our algorithm achieves an average performance improvement of 10.91% over ACO, 11.40% over PSO, and 10.55% over NSGA-II; in terms of system jitter, our algorithm

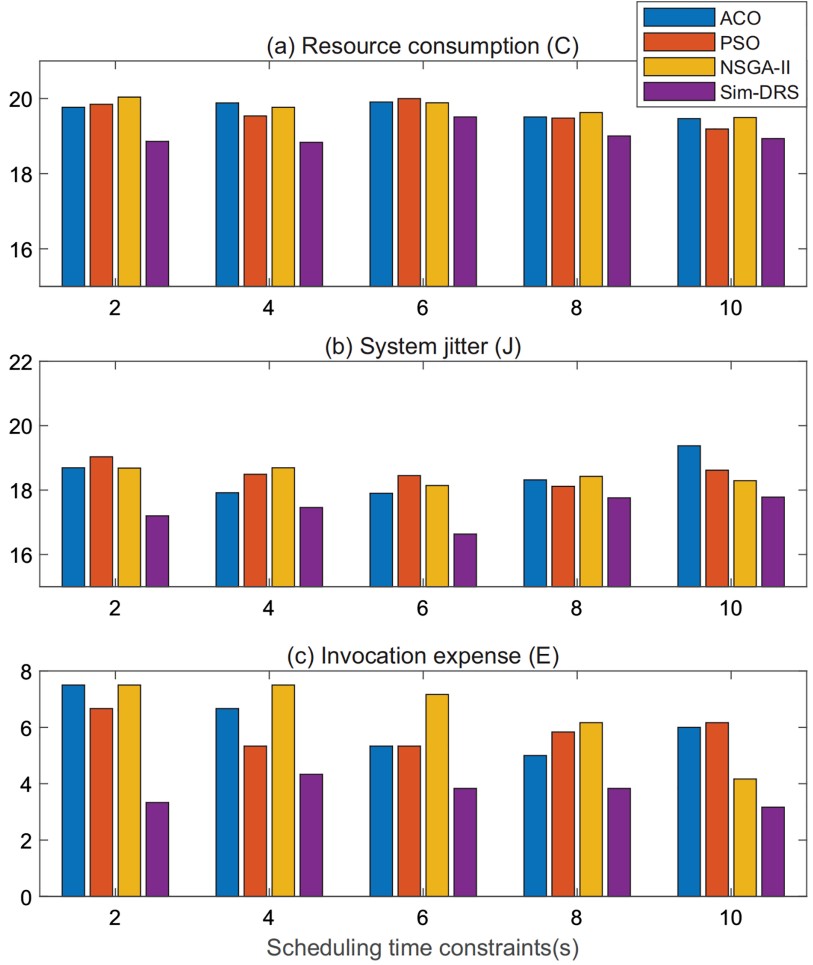

**Figure 5 Performance comparison of different algorithms under random distribution (A–C).**

achieves an average performance improvement of 42.77% over ACO, 41.53% over PSO, and 38.12% over NSGA-II; in terms of invocation expense, our algorithm achieves an average performance improvement of 9.70% over ACO, 10.58% over PSO, and 10.80% over NSGA-II. It is worth noting that Sim-DRS shows significant improvements over the other algorithms for the system jitter objective, which indicates that the solutions generated by our algorithm are more stable with a higher level of robustness of the MWS. Another important observation from Table 3 is that Sim-DRS achieves more significant improvements over the other algorithms when the time constraint is stricter (*i.e.*, tighter scheduling time). This is consistent with our similarity assumption stated in the Motivation section.

For a better illustration, we plot the performance measurements of ACO, PSO, NSGA-II, and Sim-DRS in terms of three objectives when the workload subjects to the random distribution in Figs. 5A–5C, respectively. In each figure, the *x*-axis lists the scheduling time constraint and the *y*-axis represents the measurements of the three

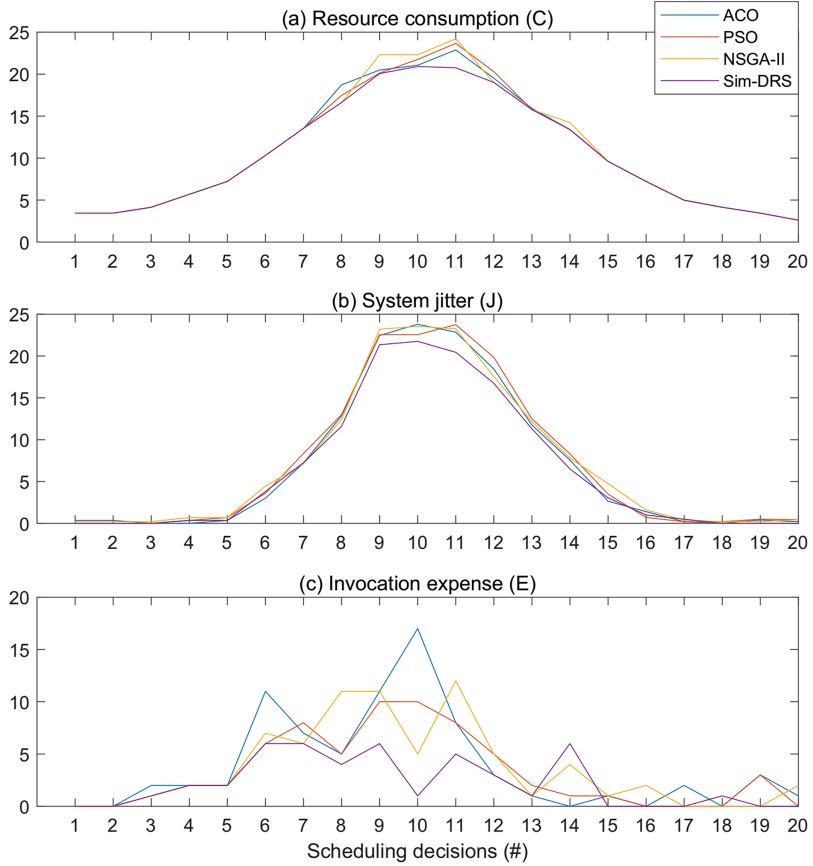

**Figure 6 Objective measurements of different algorithms with the time constraint of 2 s for 20 scheduling decisions (A–C).**

objectives. We can also conclude from Fig. 5 that Sim-DRS achieves stable improvements compared with the other three algorithms.

In summary, we can safely draw the conclusion from the experimental results that our algorithm outperforms all other algorithms in terms of three objectives, namely resource consumption, system jitter, and invocation expense, in both workloads that follow Poisson and random distribution. Note that the improvement is much significant in terms of system jitter, which means that our algorithm is more practical in the production WMS. This is because our algorithm requires less deployment efforts and is able to make the system more stable.

Figure 6 illustrates the objective measurements of different algorithms with the time constraint of 2 s for 20 scheduling decisions. We observe from Fig. 6 that Sim-DRS outperforms all other three algorithms under every scheduling decision point, followed by ACO, PSO, and NSGA-II. The difference between these algorithms is not significant at the beginning and end of the decision time points, but the difference at the middle points is. Since the workloads follow the Poisson distribution, such results indicate that Sim-DRS

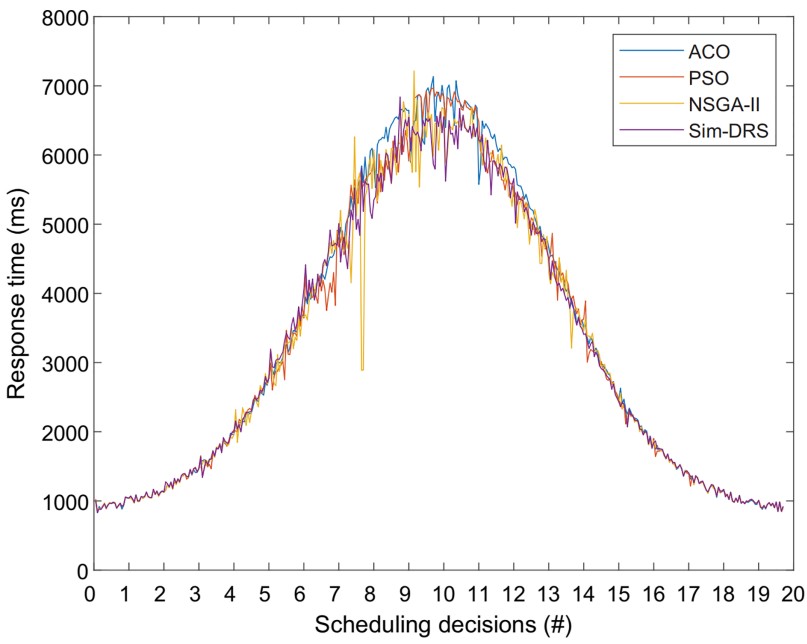

**Figure 7  Response time measurements of different algorithms with the time constraint of 2 s for 20 scheduling decisions.**

is more stable and robust under different circumstances in comparison with other algorithms.

Finally, Fig. 7 shows the response time of user requests for different algorithms with the workload following Poisson distribution. We observe from Fig. 7 that the response time measurements also follow Poisson distribution for all four algorithms, which is reasonable because there should be a positive correlation between the response time and the number of user requests. We also observed that our algorithm is better at responding to requests when the workload is heavier, which means that Sim-DRS can obtain a better performance in terms of three objectives while still ensuring a good response time.

## CONCLUSION AND FUTURE WORK

In this paper, we proposed Sim-DRS, a similarity-based dynamic resource scheduling algorithm that quickly finds promising scheduling decisions by identifying and incorporating previously-known viable solutions to similar problems as the initial population, hence obviating the need of a fresh start. We conducted extensive experiments on a well-known microservice benchmark application on disparate computing nodes on public clouds. The superiority of Sim-DRS was illustrated with various performance metrics in comparison with three state-of-the-art scheduling algorithms.

It is of our future interest to make Sim-DRS more reactive by learning previous request patterns and supporting automatic adjustments according to future workload prediction. We will also explore the possibility of using reinforcement learning-based algorithms to make more intelligent and adaptive scheduling decisions.

### Funding

This work was supported by the National Key R&D Program of China under Grant (No. 2018YFC0831200), and the National Natural Science Foundation of China under Grant (No. 61202040) with XiDian University. This work was also supported by the Key R&D Program of Shaanxi under Grant (No. 2019ZDLGY13-03-02), the Natural Science Foundation of Shannxi Province, China (2019JM-368), and the Key R&D Program of Hebei under Grant (No. 20310102D). The funders had no role in study design, data collection and analysis, decision to publish, or preparation of the manuscript.

### Grant Disclosures

The following grant information was disclosed by the authors:
National Key R&D Program of China: 2018YFC0831200.
National Natural Science Foundation of China: 61202040.
XiDian University.
Key R&D Program of Shaanxi: 2019ZDLGY13-03-02.
Natural Science Foundation of Shannxi Province, China: 2019JM-368.
Key R&D Program of Hebei: 20310102D.

### Competing Interests

Yiren Li, Pei Shen and Liang Hao are employed by HBIS Group Co., Ltd.

### Author Contributions

- Yiren Li conceived and designed the experiments, performed the computation work, authored or reviewed drafts of the paper, and approved the final draft.
- Tieke Li conceived and designed the experiments, prepared figures and/or tables, and approved the final draft.
- Pei Shen conceived and designed the experiments, prepared figures and/or tables, and approved the final draft.
- Liang Hao performed the experiments, authored or reviewed drafts of the paper, and approved the final draft.
- Wenjing Liu analyzed the data, performed the computation work, authored or reviewed drafts of the paper, and approved the final draft.
- Shuai Wang analyzed the data, prepared figures and/or tables, and approved the final draft.
- Yufei Song performed the experiments, prepared figures and/or tables, and approved the final draft.
- Liang Bao performed the computation work, authored or reviewed drafts of the paper, and approved the final draft.

## Data Availability

The data and code are available at GitHub: https://github.com/xdbdilab/Sim-DRS.

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
