# Peer review of "Sim-DRS: a similarity-based dynamic resource scheduling algorithm for microservice-based web systems"

_PeerJ Computer Science, doi:10.7717/peerj-cs.824_

## Round 0.1 · original submission · Major Revisions

The authors should address review comments to revise the manuscript.

Reviewer 1 ·

Basic reporting

This paper tackles the resource scheduling problem in web systems using a dynamic resource allocation/scheduling method, which has long been a prevalent model and commonly adopted for microservice composition in the domain of distributed computing.

The paper formulated a combinatorial optimization problem and proposed a heuristic scheduling algorithm to find feasible scheduling schemes by directly adopting existing solutions to similar problems.

Experimental design

A benchmark named TeaStore is used for experimental evaluation. TeaStore has been widely used for performance evaluation of microservice-based applications.

Validity of the findings

The performance evaluation shows that the proposed approach outperforms other methods, which is quite expected since the existing methods (which are known to be better) are used.

Additional comments

Resource scheduling in distributed computing environment is well-known to be of interest of researchers in the related area and has been investigated extensively in literatures. Although this paper includes some interesting ideas and results, the following issues need to be addressed before acceptance.

Strengths:
1. This work offers an interesting idea, i.e., solutions to similar problems often share certain structures, which is instructive for solving scheduling problems and could avoid unnecessary overhead of solving problems from scratch.
2. The performance evaluation shows positive results that make the idea worthy of further investigation and adoption.

Weaknesses:
1. A formal complexity study of the formulated problem “DRS-MWS” is missing.
2. The related work section should be better summarized, e.g., the “rule-based approach” and “heuristic approach” contains too much work in a single paragraph and these two approaches may contain some overlap and their distinction is not well described. Also, a complete refresh of the related work with more recent literatures would be appreciated.
3. Eq. 9 seems to be messed up, please correct

·

Basic reporting

no comment

Experimental design

no comment

Validity of the findings

no comment

Additional comments

This paper presents Sim-DRS, a similarity-based heuristic scheduling algorithm for microservice-based web systems that aims to find viable scheduling schemes by utilizing solutions to similar problems. More specifically, the Dynamic Resource Scheduling for Microservice-based Web Systems (DRS-MWS) is formulated as a combinatorial optimization problem, and the authors propose the Sim-DRS to solve the DRS-MWS problem. The experimental results show the superiority of Sim-DRS over three state-of-the-art scheduling algorithms.

Overall, the article is well organized and its presentation is good. The interesting idea is to obviate a fresh start in dynamic resource scheduling by incorporating previously-known viable solutions to similar problem.

However, some minor issues still need to be improved:
1. The introduction section should firstly provide an idea of the resource scheduling process before the analysis of existing methods.
2. The words in Figure 1 are not very clear, and the contents in Table 2 and Table 3 are not clearly displayed. Please improve the quality of figures and tables.
3. There are some typos and grammar errors. It is recommended that the authors should proof read the manuscript before submission.

Reviewer 3 ·

Basic reporting

The paper is easy to read and the whole process is complete and neat. More specifically, the contribution of the work consists of three key components: (1) This paper describes DRS-MWS as a combinatorial optimization problem, which clarifies the processing logic of the original problem; (2) This paper proposes Sim-DRS to solve DRS-MWS. It uses prior knowledge, that is, to directly use feasible solutions to similar problems to find a promising scheduling plan, thus effectively improving the efficiency of the algorithm; (3) This article uses a well-known microservice benchmark called TeaStore to evaluate the performance of Sim-DRS through extensive experiments under different scheduling time constraints.

Experimental design

The Sim-DRS works well in experiments, and the research results obtained should be very effective, very useful to practitioners and to promote future research directions..

Validity of the findings

This paper proposes Sim-DRS, which uses the identification and combination of previously known feasible solutions to similar problems as initial values, and uses evolutionary algorithms to quickly find promising decision-making solutions for dynamic resource scheduling.

Additional comments

To improve the paper, I have some comments though:
1. There are some formatting issues in the paper. For example, the in formula (6) should be left aligned; the result ( ) in formula 9 can try to give the reader.
2. Some symbols are not clearly defined. For example, the " " in equation (8) is unclear. If it is the dot product of the matrix, it is recommended to use " ", if it is the outer product or Hadamard product, it is also recommended to use the corresponding mainstream symbols to express; it is recommended to consider replacing (in line 180) with for more clarity. Similar issues are also recommended to be modified, etc.
3. Unify the expression of images in the text, such as "Figure 3" in line 233 and "Fig. 5" in line 330.

---

## Round 0.2 · accepted · Accept

Reviewers recommended accepting the manuscript.

Reviewer 1 ·

Basic reporting

My previous comments have been addressed in this version. I do not have any further comments.

Experimental design

My previous comments have been addressed in this version. I do not have any further comments.

Validity of the findings

My previous comments have been addressed in this version. I do not have any further comments.

Additional comments

My previous comments have been addressed in this version. I do not have any further comments.

Reviewer 3 ·

Basic reporting

No

Experimental design

No

Validity of the findings

No

Additional comments

No